

**Research Paper**
**Detecting climatically driven phylogenetic and morphological**
**divergence among spruce species (*Picea*) worldwide**
Guo-Hong Wang[1][*][#], He Li[1][#], Hai-Wei Zhao[1, 2], Wei-Kang Zhang[1]
[1] State Key Laboratory of Vegetation and Environmental Change, Institute of Botany,
Chinese Academy of Sciences, Beijing 100093, China
[2] University of the Chinese Academy of Sciences, Beijing 100049, China
#These authors contributed equally to this paper and should be regarded as co-first
authors.
[*]Address for correspondence: State Key Laboratory of Vegetation and Environmental
Change, Institute of Botany, Chinese Academy of Sciences, No. 20 Nanxincun,
Xiangshan, Beijing 100093, China
**Tel:** +86-010-6283-6585
**Fax:** +86-010-6259-0833
**E-mail:** ghwangaq@ibcas.ac.cn



**Abstract**
This study aimed to elucidate the relationship between climate and the phylogenetic
and morphological divergence of global spruces (*Picea*) in the Northern Hemisphere.
Bioclimatic and georeferenced data were collected from a total of 3388 sites
distributed within the global domain of spruce species. A phylogenetic tree and a
morphological tree for the global spruces were reconstructed based on DNA
sequences and morphological characteristics. The spatial evolutionary and ecological
vicariance analysis (SEEVA) method was used to detect the ecological divergence
among spruces. A divergence index ($D$) with (0, 1) scaling was calculated for each
bioclimatic factor at each node for both trees. The annual mean values, extreme values
and annual range of the climatic variables were among the major determinants for
spruce divergence. The ecological divergence was significant ($P<0.0016$) for 185 of
the 279 comparisons at 31 nodes in the phylogenetic tree, and for 196 of the 288
comparisons at 32 nodes in the morphological tree. Temperature parameters
($D_{max}=0.26*$ represents the annual temperature range) and precipitation parameters
($D_{max}=0.54*$ represents the precipitation of the wettest month) tended to be the main
driving factors for the primary divergence of spruce phylogeny and morphology,
respectively. The ecological divergence for the remaining splits in both trees varied
according to the sister groups or species. Generally, the $D_{max}$ of the climatic variables
was smaller in the basal nodes than in the remaining nodes. Overall, the climatic data
extracted from current spruce locations captured the ecological divergence among
spruces. In addition, the magnitude of ecological divergence among sister groups
tended to increase from the basal (older) nodes to the terminal (younger) nodes on the
phylogeny. The primary divergence of morphology and phylogeny among the
investigated spruces tended to be driven by different selective pressures. Nevertheless,
less patterning in ecological divergence was observed for the remaining splits, which
indicates that further investigations that address the geographical vicariance,
divergence and convergent evolution of spruce species are needed to determine the
forces underlying ecological divergence among sister groups or species of spruce.





**Keywords**
natural selection, niche conservatism, parallel evolution, precipitation, speciation,
temperature



## 1 Introduction

Environmental conditions play an important role in speciation (Mayr, 1947; Darnell and Dillon, 1970; Wiens, 2004; Givnish, 2010; Schemske, 2010). However, quantitative investigations of environmental influences on the origin and divergence of species are less common than expected, especially in plants (Givnish, 2010). For example, although taxonomic and phylogenetic studies have explicitly addressed phylogenetic and morphological divergence among spruces (Farjón, 1990; Sigurgeirsson and Szmidt, 1993; Fu et al., 1999; Ran et al., 2006; Li et al., 2010; Lockwood et al., 2013), ecological differentiation among sister groups or species remains unknown. Ecological vicariance differs from geographical vicariance (Wiley, 1988) and indicates the ecological differentiation among sister groups or sister species within taxa, which provides important information and ecological interpretations for the phylogenetic and morphological divergence among taxa (Escudero et al., 2009; Struwe et al., 2011).

Spruce (*Picea* A. Dietrich) is an important component of boreal vegetation and subalpine coniferous forests and has a wide geographical range that covers the northern hemisphere and extends from the Eurasian continent to North America (Farjón, 2001; Spribille and Chytry, 2002). Nearly 34 species are recognized in the genus *Picea* worldwide (Farjón, 2001). Although taxonomic schemes of *Picea* based on morphological characteristics differ slightly among authors, a consensus has been reached for the criterion to determine the first several subdivisions (Liu, 1982; Farjón, 1990; Taylor, 1993; Fu et al., 1999). Accordingly, several sections within *Picea* have



been classified based on morphological similarity. For example, section *Picea* and
section *Casicta* are characterized by quadrangular leaves and flattened leaves,
respectively (Farjón, 1990). Alternatively, spruce species can be classified into
phylogenetically distinct clades, namely clade-1, a Eurasian clade; clade-2, a North
American clade; and clade-3, an Asian clade with one North American species (Ran
et al., 2006; Lockwood et al., 2013). These chloroplast DNA sequence data-based
classification schemes have the potential to reveal the phylogenetic affinity among
spruces. We aimed to elucidate the ecological differentiations between sister groups
or species identified based on their phylogenetic affinity and morphological similarity.

A species' ecological niche depends on both the species' adaptation to its present

habitat and the legacy of its ancestors (Wiens, 2004). Although species tend to retain
similar ecological niches as their immediate ancestors in a process called phylogenetic
niche conservatism, natural selection of ecologically important traits is the key
process that determines the successful adaptation of incipient species (Peterson et al.,
1999; Webb et al., 2002; Wiens and Graham, 2005). In addition, speciation tends to
occur in geographic dimensions, whereas ecological differences evolve over time
(Peterson et al., 1999). Thus, there should be tradeoff between niche conservatism and
ecological differences among splits in the phylogeny of given taxa over evolutionary
time scales. Spruces likely originated in the early Tertiary or late Cretaceous era. The
fossil spruce species *Picea burtonii* Klymiuk et Stockey is regarded as the earliest
fossil record for *Picea* and dates to approximately 136 Ma (Klymiuk and Stockey,
2012). The ancestor of extant spruces dates to the Oligocene (Sigurgeirsson and



Szmidt, 1993; LePage, 2001; Ran et al., 2006; Lockwood et al., 2013). The
divergence times of extant spruces occurred over a long time scale, with a range of
approximately 28 Ma to several Ma from the basal node to the end nodes (Lockwood
et al., 2013). We hypothesize that there should be a relationship between the time
since separation and the magnitude of ecological divergence or niche conservatism.
Specifically, we expect to observe an increasing magnitude in terms of ecological
divergence among sister groups from the basal nodes (older) to the end nodes
(younger) on the evolutionary time scales because natural selection would favor
species with high levels of ecological adaptation.
Although phylogenetically close species are likely to be similar in appearance to
one another, differences in the rate of evolution may substantially obscure these
similarities (Baum et al., 2005). In the genus *Picea*, none of the morphology-based
classification schemes are congruent with or supported by the schemes derived from
cpDNA-based phylogenies. Therefore, spruce species within a taxonomic section are
not necessarily more similar in phylogenetic relatedness than those between sections
or subsections, which indicates that parallel evolution, i.e., the repeated appearance of
similar characteristics that occur among distantly related species (Went, 1971;
Hoekstra and Price, 2004; Schluter et al., 2004; Orr, 2005), occurs in *Picea*. Therefore,
we hypothesize that the divergence of morphology and phylogeny among the
investigated spruce species may be subject to different selective pressures under
parallel evolution.
Evolutionary trees indicate historical relationships among organisms (Baum et al.,





2005). This "tree-thinking" approach has been used in almost all branches of biology
to detect relatedness among organisms (Baum and Offner, 2008) and to examine
ecological divergence between sister clades or species (Struwe et al., 2011). In this
study, tree-thinking methods were used to examine the ecological divergence among
spruce species worldwide by reconstructing a phylogenetic tree and a morphological
tree. A dataset of spruce species was compiled to test our hypothesis by answering the
following three questions: are the climatic variables extracted from the current spruce
locations correlated with the divergence among spruces? If so, is there a relationship
between the time since separation and the magnitude of ecological divergence? Lastly,
is the morphological and phylogenetic divergence among spruce species subject to
different selective pressures?
**2 Materials and Methods**
**2.1 Distribution data**
The sampling sites were selected from within the entire natural range of spruce
species in the Northern Hemisphere (latitude: 22.8-69.9°N; longitude: 53-165°W,
5-155°E; altitude: 103-4700 m a.s.l., Figure 1). Between 34 and 35 species are
included in the genus *Picea* (Farjón, 2001). The global spruce checklist used in this
study was primarily based on Farjón (2001) but refined according to the flora of
China (Fu et al., 1999). Specifically, because two species distributed in western China
according to Farjón (1990), *Picea retroflexa* and *P. aurantiaca*, were treated as a
synonym and a variety of *P. asperata,* respectively, in the flora of China, we followed
the Chinese classification. Accordingly, the checklist used for this study contained 33



spruce species.
Georeferenced data for the 33 spruce species was partially downloaded from the
Global Biodiversity Information Facility (GBIF), an international open data
infrastructure. Original data in the GBIF are derived from various sources, such as
natural history explorations (specimens or records) collected over the past 300 years,
current observations and automated monitoring programs (GBIF, 2015). We carefully
verified the original data downloaded from GBIF by excluding those data points with
geolocations outside of the natural distribution ranges (either horizontally, vertically
or both). As a result, approximately 2397 point locations from the GBIF remained
after the verification, and they primarily represented spruce species in North America,
Europe and East Asia (Japan and Korea Peninsula). Additional data for the spruce
species from the Chinese mainland and Taiwan (approximately 991 locations for 16
species) were obtained from geo-referenced herbarium collection records
(approximately 490) (Li et al., 2016) from the herbarium of the Institute of Botany,
Chinese Academy of Sciences; recent fieldwork (approximately 370 sites,
unpublished); and published sources (approximately 41 sites) (Tseng, 1991; Yang et
al., 2002). As a result, 3388 point locations for the 33 spruce species were available
for this analysis.
**2.2 Climatic variables**
A total of 19 bioclimatic variables with a resolution of approximately 1 km$^2$ for the
3388 point locations were acquired and downloaded from WorldClim V. 1.4
(http://www.worldclim.org) (Hijmans et al., 2005). These variables included annual



mean temperature, mean temperature diurnal range, isothermality, temperature
seasonality, maximum temperature of the warmest month, minimum temperature of
coldest month, annual temperature range, mean temperature of the wettest quarter,
mean temperature of the driest quarter, mean temperature of the warmest quarter,
mean temperature of the coldest quarter, annual precipitation, precipitation of the
wettest month, precipitation of the driest month, precipitation seasonality,
precipitation of the wettest quarter, precipitation of the driest quarter, precipitation of
the warmest quarter and precipitation of the coldest quarter. The values of each
climate variable at each site were extracted using the software QGIS
(http://qgis.osgeo.org), and the final data were exported to an Excel worksheet for
subsequent analysis. A factor analysis was conducted to eliminate the redundant
climatic variables, and a principal component analysis (PCA) of the climatic variables
was performed using the SPSS statistical package (SPSS, Chicago, IL, USA).
**2.3 Data analysis**
DNA sequences were retrieved from the NCBI GenBank (www.ncbi.nlm.nih.gov) to
reconstruct a phylogenetic tree of the 33 spruce species (Figure 2). This phylogenetic
tree was constructed based on 3 plastid (trnL-trnF, trn-psbA, and trnS-trnG) and 2
mitochondrial (nad5 intron1 and nad1 intron 2) DNA sequences, and it was equivalent
to that of Lockwood et al. (2013), who proposed an improved phylogeny of *Picea*.
In addition, we reconstructed a morphological tree of the 33 spruce species (Figure
3) based on Farjón (1990), Taylor (1993), and Fu et al. (1999). The first several splits
in the tree primarily revealed divergence in the shape of the leaf cross section, the



position of the stomatal line on the leaf surface, and the texture and arrangement of
the seed scale, whereas traits such as the size of the leaf, seed cone and seed scale and
the hairiness of the leaf or twig are important indicators for subsequent splits in the
trees. To detect ecological divergence among sister groups or species in the
above-mentioned trees, we used the spatial evolutionary and ecological vicariance
analysis (SEEVA, Struwe et al., 2011), which can incorporate bioclimatic data with
phylogenetic data and morphological data using statistical methods to investigate
ecological vicariance in speciation. We constructed a morphological tree and
phylogenetic tree that contained 32 and 31 nodes, respectively. The SEEVA compares
the differences between each of the bioclimatic variables for each node. A divergence
index ($D$) with (0,1) scaling was calculated for each bioclimatic factor at each node.
$D$=0 indicates no difference between sister clades or groups, whereas $D$=1 indicates a
maximum difference. Fisher's exact test (Fisher, 1958), which generally provides a
better $P$-value for tests with small sample sizes, was performed to determine the
significance of $D$. Because 31 and 32 independent tests were conducted for each of
the bioclimatic variables, a $P$-value less than 0.0016 indicated a significant difference
in the ecological features for splits at a given node after performing a Bonferroni
correction, i.e., $\alpha$=0.05/31 or 32≈0.0016. Details on the calculations are available in
Struwe et al. (2011). The SEEVA software can be downloaded from
http://seeva.heiberg.se. In addition, we compared the means of the 9 abiotic variables
among sister groups at several key splits (i.e., the first two split levels) of both
constructed trees using a one-way analysis of variance (ANOVA) to further interpret



the observed ecological divergence.
**3 Results**
**3.1 Variation in climatic variables**
A factor analysis of the bioclimatic variables across sampling sites revealed five
dominant climatic gradients that accounted for 94.1% of the variance (Table 1). The
first component, which had an eigenvalue of 8.27 and accounted for 29.8% of the
variance, was a gradient characterized by variation in temperature variables. The
second component, which had an eigenvalue of 3.59 and accounted for 21.6% of the
variance, was a gradient characterized by variation in precipitation variables. The
third, fourth and fifth components, which accounted for 19.1%, 14.4% and 9.1% of
the variance, respectively, were characterized by variation in the precipitation of the
driest month or quarter and precipitation seasonality; maximum temperature of the
warmest month or quarter; and mean temperature of the wettest quarter and
precipitation of the coldest quarter, respectively. Therefore, we selected eight
bioclimatic variables for subsequent analysis, including four temperature variables
(annual mean temperature, minimum temperature of the coldest month, maximum
temperature of the warmest month and annual temperature range) and four
precipitation variables (annual precipitation, precipitation of the wettest month,
precipitation of the driest month and precipitation of the coldest quarter). In addition,
elevation as a spatial variable was also used to detect the ecological vicariance among
sister groups because spruce is an elevation-sensitive taxa, which is represented its
geographical distribution (Farjón, 1990; Taylor, 1993; Fu et al., 1999).





**3.2 Ecological divergence among sister groups or species in the phylogeny of**

*Picea*

Ecological divergence as indicated by the (0, 1) scaled index of *D* was significant

(*P*<0.0016) for 185 of the 279 comparisons at 31 nodes in the phylogeny of *Picea* (see

Table S1 in Supplement S1). The first split, which yielded node-2 (clade-1) and

node-14 (clade-2 and clade-3), was significant for all 9 environmental variables. The

annual temperature range (*D*=0.26*) showed higher divergence, and it was followed

by elevation (*D*=0.25*) and precipitation of the driest month (*D*=0.20*). The spruce

species in clade-1 tended to occur in climates with a lower annual temperature range

and lower precipitation compared with the spruce species in node-14. The divergence

within node-14 and between node-15 (clade-2) and node-22 (clade-3) was also

significant for all 9 environmental variables. The parameters precipitation of the

coldest quarter, precipitation of the driest month and precipitation of the wettest

month had relatively high divergence (*D*=0.66* to 0.42*), elevation exhibited

substantial divergence (*D*=0.46*), whereas the temperature variables showed lower

divergence (*D*=0.13* to 0.31*). Compared with clade-3, clade-2 occurred in climates

with lower precipitation levels and a higher annual temperature range. Node-2

represented a split within clade-1 (the Eurasian clade) between a subclade at a higher

elevational zone (in Caucasian area and Japan) with a warmer and wetter climate and

a subclade at a lower elevational zone (esp. in boreal area) with a cold and dry climate.

The elevation and temperature features showed relatively higher divergence (*D*=0.17*

to 0.38*) compared with the precipitation variables (*D*=0.03* to 0.23*) (Figure 2,



Table 2).
The ecological divergence for the subsequent 28 splits in the phylogeny of *Picea*,
i.e., from node-3 to node-13 and from node-15 to node-31, was significant for
approximately 63% of the comparisons. However, a universal pattern was not
observed in terms of the ecological divergence for the remaining splits, which varied
according to the sister groups or species. This finding suggests that a particular
combination of environmental features is important for particular splits among sister
groups or species (Figure 2, Table 2).
**3.3 Ecological divergence among sister groups or species in the morphology of**
***Picea***
Ecological divergence was significant ($P<0.001$) for 196 of the 288 comparisons at
32 nodes in the morphology tree of *Picea* (see Table S2 in Supplement S1). Of the 32
nodes, we focused on three splits that represent several key morphological divergence
in *Picea*. Specifically, the split of node-1 represents divergence in the shape of the
leaf cross section and the position of the stomatal line on the leaf surface, whereas the
split of node-2 and node-25 represents divergence in the texture and seed scale
arrangement. The remaining 29 splits, i.e., from node-3 to node-24 and from node-26
to node-32, reflect divergence in the leaf size, seed cone size, hairiness (pubescent vs.
glabrous) and branchlet color, and these differences were significant for
approximately 65% of the comparisons (Figure 3).
**3.4 Ecological divergence of the leaf cross section: quadrangular vs. flattened**
The first split of the morphology-defined topology tree (Figure 3) yielded node-2 (leaf



quadrangular) and node-25 (leaf flattened) and was significant for all 9 environmental
variables. Precipitation features ($D$=0.16\*-0.54\*), predominantly precipitation of the
wettest month, showed much stronger divergence compared with that of temperature
features ($D$=0.05\*-0.18\*), with elevation showing a moderate divergence ($D$=0.30\*).
Spruce species with quadrangular leaves tended to be favored by drier habitats with
higher temperature annual ranges in lower elevational zones, which is inconsistent
with the habitats for spruces with flattened leaves (Table 2). Such an overall pattern,
however, does not necessarily hold true for the sister groups or species that present
different leaf cross sections (flattened vs. quadrangular) but close phylogenetic
relationships. Sister groups or species at node-10, node-13, node-18, node-26 and
node-31 in the phylogeny tree are relevant examples (Figure 3). For example,
although elevation was important for the divergence between *P. jezoensis* and *P.*
*glehnii* (node-10), temperature parameters were important for the divergence between
*P. wilsonii* and *P. purpurea* (node-31).
**3.5 Ecological divergence of seed scale: closely arranged vs. loosely arranged**
The second-level splits in the morphological tree (Figure 3) yielded two pairs of
sister groups, namely node-3 vs. node-24 (within node-2) and node-26 vs. node-29
(within node-25). These two pairs of spruce sister groups collectively indicated
divergence in the seed scale characteristics, i.e., closely arranged seed scales with a
rigid woody texture vs. loosely arranged seed scales with a thin, flexible, leathery or
papery texture. For the split within node-2, elevation showed the highest divergence
($D$=0.51\*) and was followed by annual temperature range ($D$=0.48\*) and precipitation



of the driest month (*D*=0.35\*), whereas the remaining climatic variables had
significant but relative low divergence (*D*=0.06\*-0.25\*). Compared with the results
for node-24 (loosely arranged seed scales), the species in node-3 (closely arranged
seed scales) tended to occur in lower elevational zones with higher precipitation of the
driest month and a wider variation of annual temperature range (Table 2). For the split
within node-25, both the minimum temperature of the coldest month (*D*=0.46\*) and
precipitation of the driest month (*D*=0.43\*) showed substantial divergence, with a
moderate divergence for elevation (*D*=0.35\*). Compared with the results for node-26
(loosely arranged seed scales), the species in node-29 (closely arranged seed scales)
tended to occur in lower elevational zones with higher temperature and greater
precipitation in the coldest quarter (Table 2).
**3.6 Magnitude of ecological divergence and time since separation**
Nine levels of splits occurred in the phylogenetic tree. From level 1 to 3, the (0,1)
scaled index of divergence (*D*) tended to increase in terms of the median value,
maximum value and interquartile range. From level 3 to 9, the maximum value of *D*
for most cases (except level 8) was approximately 1, whereas the median and the
interquartile range were less structured (Figure 4a). There were 10 levels of splits in
the morphological tree. The maximum value of *D*, which was even slightly higher for
level 1 (*D*=0.54) than level 2 (*D*=0.48), was approximately 1 for the remaining levels.
The median tended to increase from level 1 to 7 and then decrease from level 7 to 10.
The interquartile range tended to increase from level 1 to 9 (Figure 4b).
**4 Discussion**



**4.1 Climatic data extracted from current spruce locations captures the ecological divergence among spruces**

In this study, we used climatic data extracted from the current locations of spruce populations to examine the ecological divergence among spruce species at various time scales from approximately 28 Ma to several Ma. Our results showed significant divergence for the ecological niches among sister groups throughout the phylogenetic tree and the morphological tree, which indicated the overall relevance of the climatic data on spruce ecological divergence at various time scales. However, the magnitude of ecological divergence (as indicated by the divergence index ($D$)) decreased with the time since the separation of species and became much more specific, i.e., variation of $D$ among the nine environmental variables was larger in the more recent splits than in the basal splits.

This finding is likely associated with the incompatibility of the time scale between environmental data and ecological divergence because the environmental data extracted from the current locations tended to be more relevant to the divergence of younger nodes than older nodes. The low ecological divergence observed at the first split in both trees should be an indicator of high ecological niche conservatism (Struwe et al., 2011). Thus, the higher divergence observed for the younger sister groups or sister species might suggest a strong selective effect of climate on extant spruce species derived from more recent splits; however, the observed pattern is likely related to the strong species interactions that obscure the splits at the basal or first several nodes and the fewer species and therefore relatively more simple trait



composition and weak interactions of the sister groups or species within each node in
the more recent splits.
Exceptions to the above-mentioned trend were observed for a few sister groups or
species in the phylogenetic tree. Specifically, within clade-3, significant ecological
divergence was not detected for the split (node-29 in Figure 2) between *P. spinulosa*
and *P. brachytyla*. These two sister species are distributed in the Circum-Tibetan
Plateau and their geographical ranges are adjacent. *P. spinulosa* is distributed in the
Mt. Himalaya region and has a narrow range (S Xizang, Bhutan, Nepal and Sikkim),
whereas *P. brachytyla* is distributed in the SE to E Tibetan Plateau and has a wide
range. These differences suggest that instead of ecological divergence, geographical
isolation caused by the deep valleys and high mountain peaks in this area, which act
as barriers to gene flow between species, might have played a major role in the
speciation of these two sister species (Li et al., 2010). Nevertheless, we cannot rule
out the possibility that the selected climate parameters do not adequately describe the
climatic determinants of spruce distributions. Our first hypothesis is largely verified
by the findings of our study and those of a previous case study (Struwe et al., 2011).
**4.2 Temperature features tend to be the main driving factors of the primary**
**divergence of spruce phylogeny**
Of the 31 splits in the phylogeny tree of *Picea*, the first split is much more
important than the subsequent splits because it represents "the primary trigger" that
led to the divergence of the genus. Temperature parameters showed higher divergence
for the first split of the spruce phylogeny, although moisture factors were not





negligible. The first split of the spruce phylogeny occurred at approximately 28 Ma in
a period with severe oscillations of global temperature, which sharply declined at the
end of the Eocene and then warmed during the late Oligocene and early Miocene
(Lockwood et al., 2013). This oscillation may provide an explanation for the higher
divergence of temperature features. The divergence among the nine environmental
variables for the subsequent splits, however, varied according to the sister groups or
species.
It is well established that the variations in the historical climate associated with the
advancement and retreat of ice sheets during the late Tertiary and Quaternary periods
played an important role in determining plant distributions (Walker, 1986; Hewitt,
2000). In this process, old taxa became extinct or survived in refugia, whereas derived
taxa dispersed to new locations and underwent severe selection by climate (Hewitt,
2000; Hampe and Petit, 2010). Therefore, the formation of biogeographical plant
patterns is a product of interactions among these processes (Wolf et al., 2001).
In fact, considerable variations in geology and climate have occurred since spruce
originated in the late Oligocene. For example, the earliest spruce pollen fossil is from
the late Oligocene to the early Miocene in Asia and was found on the Tibetan Plateau
(Wu et al., 2007), and spruce pollen has frequently been found in sediments
originating from the late Pliocene and the Pleistocene in northern, eastern and
southwestern China (Xu et al., 1973; Xu et al., 1980; Shi, 1996) and Taiwan (Tsukada,
1966). A higher proportion of spruce pollen in specific sediments is generally
assumed to indicate a cold period, whereas a lower proportion of spruce pollen





indicates a warmer period (Xu et al., 1980). The proportion of spruce pollen in the
sediments varied substantially with the geological age of the sediments, suggesting
that spruce underwent frequent expansion and retreat during glacial cycles. In North
America, fossils of Brewer spruce (*P. breveriana*) have been observed in northeastern
Oregon in Miocene deposits that date to more than 15 Ma years ago; however, the
present distribution of Brewer spruce is different from the distribution of the fossil
locations, indicating that expansion and retreat occurred in the past (Waring et al.,
1975). It is difficult to match all the details of paleo-geological or paleo-climatic
events to the ecological divergence observed for specific nodes, although our findings
offer a quantitative interpretation with respect to the influence of climate on spruce
speciation.
**4.3 Precipitation features tend to be the main driving factors of the primary**
**divergence of spruce morphology**

The morphological tree in this study was based on spruce taxonomic schemes and

highlights the divergence between leaf cross sections in spruce. Although this
morphological tree is an artificial scheme, our results indicate that precipitation
features were "the primary trigger" of the divergence between quadrangular leaves
and flattened leaves among spruce species. A universal pattern was not observed for
the climatic variables with respect to the ecological divergence of spruce morphology,
which varied according to the specific nodes or splits.

The first split of the basal node of the morphological tree was based on the leaf

cross section (i.e., quadrangular vs. flattened); however, each sister group is actually a



combination of multiple traits, including the size, shape, color and pubescent/glabrous
state of the seed cones, seed scales, bud scales, leaves, leaf apex, and first- and
second-year branchlets (Farjón, 1990; Sigurgeirsson and Szmidt, 1993; Fu et al.,
1999). The morphological and morphometric traits of spruce species have been
demonstrated to produce strong climatic signals; however, specific traits for different
species do not necessarily exhibit the same response to specific environmental
gradients (Wang et al., 2015; Li et al., 2016). This inconsistency in response is likely
due to parallel evolution because morphological similarity among species does not
necessarily coincide with the phylogenetic relatedness of species (Went, 1971; Orr,
2005). Accordingly, spruce species with similar morphological characteristics but
distant phylogenetic relatedness may differ because of the tradeoff between niche
conservatism and ecological divergence. In addition, the composition of traits within a
species is also species specific. For example, the shape of the leaf cross section
co-varies along with the stomatal line position on the leaf surface, seed scale
arrangement and seed scale texture. However, evidence in support of the co-evolution
between the leaf cross section (quadrangular (Q) vs. flattened (F)) and seed scale
arrangement (closely (C) vs. loosely (L)) has not been observed. Trait combinations
such as Q+C, Q+L, F+C and F+L are found in 22, 2, 4 and 5 of the 33 species in
*Picea,* respectively (Farjón, 2001). Therefore, without providing additional details, a
universal pattern of ecological divergence cannot be predicted for the entire
morphological tree of *Picea*.
**4.4 Divergence of morphology and phylogeny among spruce species is affected by**



**different selective pressures under parallel evolution**

Closely related species in a phylogenetic tree tend to be similar in appearance, although this may not be so under parallel evolution (Hoekstra and Price, 2004; Baum et al., 2005; Orr, 2005), and both cases can be observed in spruce. First, of the three clades in the phylogenetic tree, most of the spruce species (19 of 22) in clade-1 and clade-2 tended to have quadrangular leaves, whereas nearly half of the spruce species (6 of 11) in clade-3 tended to have flattened leaves. In addition, two North American species, *P. rubens* and *P. mariana*, are sister species in both constructed trees. Accordingly, the morphological divergence and phylogenetic divergence of these species are subject to the same selective pressures. Second, cases of parallel evolution are quite obvious. For example, two Asian species, *P. purpurea* and *P. wilsonii*, are sister species in the phylogenetic tree but are located in different sections of the morphological tree; this scenario is also observed for another two North American species, *P. glauca* and *P. engelmannii*. As a result, the morphological and phylogenetic divergences for these species pairs are subject to different selective pressures, which suggests that the divergence of morphology and phylogeny among the species in question may or may not be subject to different selective pressures depending on the process of speciation.

**5 Summary and conclusions**

In summary, the influence of climate on the divergence of the morphology and phylogeny of spruces is mediated by a number of biotic and abiotic factors, such as geographical isolation, niche conservatism and ecological adaptation. A major finding from this study is that temperature and precipitation parameters tended to be the main





driving factors for the primary divergence of spruce phylogeny and morphology,
respectively. Our hypotheses are largely verified by the findings of the present study.
However, exceptions to the overall pattern cannot be ignored. For example, although
most spruce species with quadrangular leaves tend to occur in drier habitats, Taiwan
spruce (*P. morrisonicola*) presents quadrangular leaves and is naturally distributed in
subtropical areas with abundant rainfall; thus, its present distribution is likely within a
refugium from the postglacial period (Tsukada, 1966; Xu et al., 1980). Further work
that considers all of the determinants is required to understand the forces driving
ecological divergence among spruce sister groups or species.
**6 Data availability**
The relevant data are within the paper and its Supporting Information files.
**7 Author contribution**
GHW conceived and designed the experiments. All authors performed the
experiment. GHW and HL analyzed and interpreted the data. All authors wrote the
paper and declare they have no conflict of interest.
**8 Acknowledgements**
We thank Xing Bai, Lijiang Zhou, Miao Ma, Qinggui Wang, Hongchun Wang, Zhi
Ma, Ziying Chen and Tiancai Chen for providing field assistance. This work was
supported by National Natural Science Foundation of China (41571045), the Chinese
National Basic Research Program (2014CB954201), and the National Natural Science
Foundation of China (30870398).

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



**Table 1.** Factor analysis showing the eigenvalues, variance percentages, cumulative
percentages and correlations of 19 bioclimatic variables with the five components.
Bioclimatic variables in bold were selected for further analysis.

| Bioclimatic variables | | Components | | | | |
|---|---|---|---|---|---|---|
| | | 1 | 2 | 3 | 4 | 5 |
| | Eigenvalues | 8.27 | 3.60 | 2.51 | 2.26 | 1.24 |
| | Variance % | 43.52 | 18.93 | 13.21 | 11.89 | 6.51 |
| | Cumulative % | 43.52 | 62.46 | 75.67 | 87.55 | 94.06 |
| **Annual Mean Temperature (AMT)** | | **0.803** | 0.222 | 0.082 | 0.513 | -0.152 |
| Mean Diurnal Range | | -0.118 | -0.155 | -0.686 | 0.476 | 0.31 |
| Isothermality | | 0.687 | 0.283 | -0.45 | 0.158 | 0.307 |
| Temperature Seasonality | | -0.928 | -0.237 | -0.099 | 0.204 | -0.12 |
| **Max Temperature of Warmest Month (MTWM)** | | 0.037 | -0.155 | -0.129 | **0.968** | 0.01 |
| **Min Temperature of Coldest Month (MCM)** | | **0.931** | 0.216 | 0.257 | 0.086 | 0.006 |
| **Temperature Annual Range (TAR)** | | **-0.854** | -0.267 | -0.294 | 0.329 | -0.001 |
| Mean Temperature of Wettest Quarter | | -0.123 | 0.091 | -0.066 | 0.48 | -0.788 |
| Mean Temperature of Driest Quarter | | 0.841 | 0.093 | 0.138 | 0.116 | 0.408 |
| Mean Temperature of Warmest Quarter | | 0.14 | 0.02 | 0.04 | 0.918 | -0.294 |
| Mean Temperature of Coldest Quarter | | 0.946 | 0.24 | 0.108 | 0.179 | 0.007 |
| **Annual Precipitation (AP)** | | 0.306 | **0.856** | 0.365 | -0.041 | 0.178 |
| **Precipitation of Wettest Month (PWM)** | | 0.288 | **0.942** | -0.006 | -0.033 | 0.109 |
| **Precipitation of Driest Month (PDM)** | | 0.147 | 0.255 | **0.911** | 0.008 | 0.087 |
| Precipitation Seasonality | | -0.109 | 0.255 | -0.887 | -0.006 | -0.131 |
| Precipitation of Wettest Quarter | | 0.297 | 0.937 | 0.026 | -0.038 | 0.134 |
| Precipitation of Driest Quarter | | 0.175 | 0.302 | 0.894 | -0.003 | 0.152 |
| Precipitation of Warmest Quarter | | 0.144 | 0.888 | 0.086 | -0.057 | -0.313 |
| **Precipitation of Coldest Quarter (PCQ)** | | 0.323 | 0.402 | 0.418 | -0.016 | **0.652** |



**Table 2.** Mean comparisons of the elevation and 8 bioclimatic variables (mean ± SD,
abbreviations are the same as in Table 1) between sister groups at the first two split
levels of both the phylogeny tree and the morphology tree. Mean ± SD values marked
with different letters indicate a significant difference at *P*<0.05, and the same letter
indicates a non-significant difference (*P*>0.05).

| | *N* | Elevation (m) | AMT (℃) | MTWM (℃) | MTCM (℃) | TAR (℃) | AP (mm) | PWM (mm) | PDM (mm) | PCQ (mm) |
|---|---|---|---|---|---|---|---|---|---|---|
| **Phylogeny Nodes** | | | | | | | | | | |
| **Sister Groups:** node-2 (clade-1) vs. node-14 (clade-2 + clade-3) | | | | | | | | | | |
| 2 | **1568** | **964±750** [a] | **3.2±4.2** [a] | **19.6±3.7** [a] | **-12.6±8.6** [a] | **32.1±9.5** [a] | **845.8±416.9** [a] | **117.1±52.3** [a] | **38.0±25.7** [a] | **158.9±124.2** [a] |
| 14 | **1820** | **1721±1150** [b] | **3.8±5.0** [b] | **21.8±3.9** [b] | **-13.9±8.8** [b] | **35.7±8.8** [b] | **910.7±727.6** [b] | **143.6±119.0** [b] | **26.9±27.8** [b] | **186.5±209.3** [b] |
| **Sister Groups:** node-15 (clade-2) vs. node-22 (clade-3) | | | | | | | | | | |
| 15 | 1100 | **1176±906** [a] | **2.5±5.0** [a] | **22.5±3.6** [a] | **-16.6±8.2** [a] | **39.1±7.3** [a] | **784.1±442.6** [a] | **106.3±61.6** [a] | **35.7±27.7** [a] | 190.7±180.0 [a] |
| 22 | 720 | **2554±971** [b] | **5.9±4.3** [b] | **20.6±4.0** [b] | **-9.9±8.1** [b] | **30.6±8.4** [b] | **1104.0±989.0** [b] | **200.8±157.0** [b] | **13.5±21.8** [b] | 180.0±247.4 [a] |
| **Sister Groups:** node-3 vs. node-11 (two sister groups within clade-2) | | | | | | | | | | |
| 3 | 1502 | **951±755** [a] | **3.0±4.2** [a] | **19.4±3.6** [a] | **-12.8±8.6** [a] | **32.2±9.7** [a] | **834.5±411.2** [a] | 116.2±51.3 [a] | **37.4±25.8** [a] | **157.2±126.0** [a] |
| 11 | 66 | **1275±542** [b] | **7.1±2.8** [b] | **22.9±2.6** [b] | **-7.5±3.7** [b] | **30.4±2.8** [b] | **1101.8±464.7** [b] | 137.8±70.0 [a] | **52.3±16.7** [b] | **196.3±63.3** [b] |
| **Morphology Nodes** | | | | | | | | | | |
| **Sister Groups:** node-2 vs. node-25 (i.e., quadrangular leaf group vs. flattened leaf group) | | | | | | | | | | |
| 2 | 2857 | **1191±915** [a] | **3.1±4.7** [a] | 20.8±4.0 [a] | **-14.0±8.8** [a] | **34.8±9.7** [a] | **849.4±624.2** [a] | **120.0±95.2** [a] | **35.3±27.2** [a] | **163.8±146.4** [a] |
| 25 | 531 | **2337±1222** [b] | **5.8±3.7** [b] | 20.7±3.7 [a] | **-9.3±6.6** [b] | **29.9±5.5** [b] | **1048.5±452.1** [b] | **192.2±67.9** [b] | **14.5±21.0** [b] | **226.8±279.7** [b] |
| **Sister Groups:** node-3 vs. node-24 (i.e., within quadrangular leaf group: seed scale closely arranged group vs. loosely arranged group) | | | | | | | | | | |





| | | | | | | | | | |
|---|---|---|---|---|---|---|---|---|---|
| 3 | 2530 | **1059±850** [a] | **3.0±4.8** [a] | **20.5±3.9** [a] | **-14.3±9.2** [a] | 34.8±10.2 [a] | **864.7±646.3** [a] | **121.6±97.8** [a] | **36.6±28.4** [a] | **155.8±135.1** [a] |
| 24 | 327 | **2219±729** [b] | **3.7±3.7** [b] | **22.8±4.0** [b] | **-12.1±4.8** [b] | 34.8±4.2 [a] | **730.9±396.0** [b] | **107.7±70.6** [b] | **25.7±10.8** [b] | **225.9±204.9** [b] |

**Sister Groups:** node-26 vs. node-29 (i.e., within flattened leaf group: seed scale closely arranged group vs. loosely arranged group)

| | | | | | | | | | |
|---|---|---|---|---|---|---|---|---|---|
| 26 | 283 | **2806±1301** [a] | **4.6±4.1** [a] | **19.0±3.3** [a] | **-12.4±7.3** [a] | **31.4±6.7** [a] | **996.1±564.2** [a] | 190.1±77.4 [a] | 15.1±23.7 [a] | **125.5±252.6** [a] |
| 29 | 248 | **1802±854** [b] | **7.2±2.5** [b] | **22.5±3.2** [b] | **-5.7±3.0** [b] | **28.2±2.9** [b] | **1108.4±261.7** [b] | 194.6±55.3 [a] | 13.8±17.4 [a] | **342.4±264.2** [b] |
| 4 | 2118 | **1124±890** [a] | 3.0±4.9 [a] | **20.0±3.9** [a] | **-13.8±9.5** [a] | **33.8±10.5** [a] | 853.8±682.2 [a] | **124.6±105.6** [a] | **33.3±26.2** [a] | **149.0±139.0** [a] |
| 21 | 412 | **724±487** [b] | 3.2±4.3 [a] | **23.2±2.9** [b] | **-17.0±6.9** [b] | **40.1±6.0** [b] | 921.0±412.3 [a] | **106.2±33.1** [b] | **53.2±33.0** [b] | **190.8±105.7** [b] |



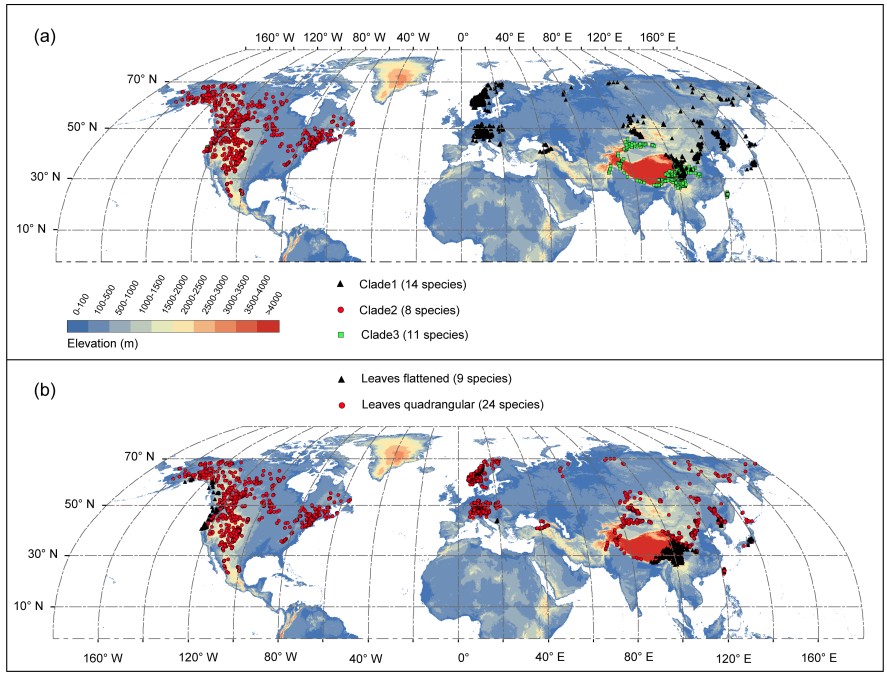

**Figure 1.** Sites were sampled across the entire range of spruces worldwide. Sites marked with different symbols represent three phylogenetically distinct clades (a), and two morphological groups (b), respectively. Elevation gradients are indicated by colored fields.





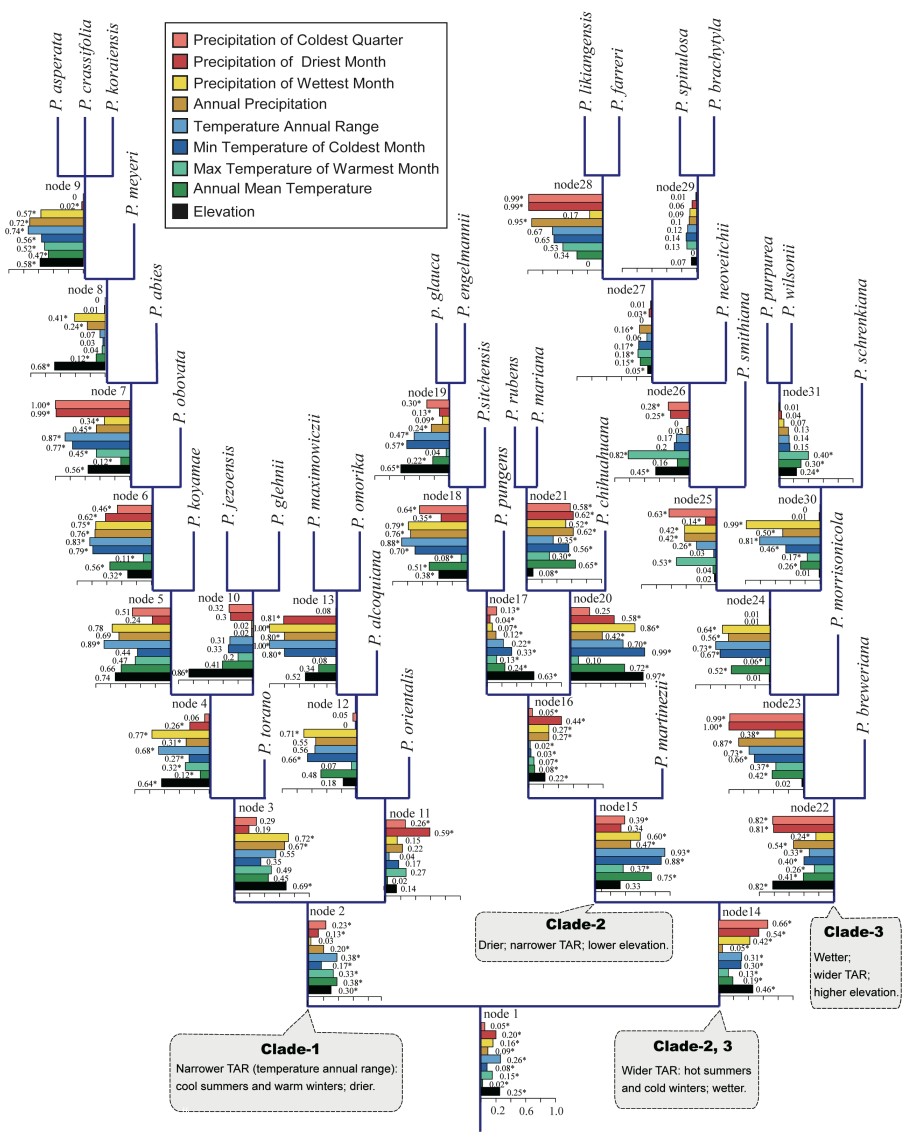

**Figure 2.** Divergence indices (scales range from 0-1) shown as histograms for elevation and for the 8 bioclimatic variables for each node of the phylogeny of *Picea* worldwide. *Indicates a significant difference in ecological features after Bonferroni correction (*P*<0.0016).





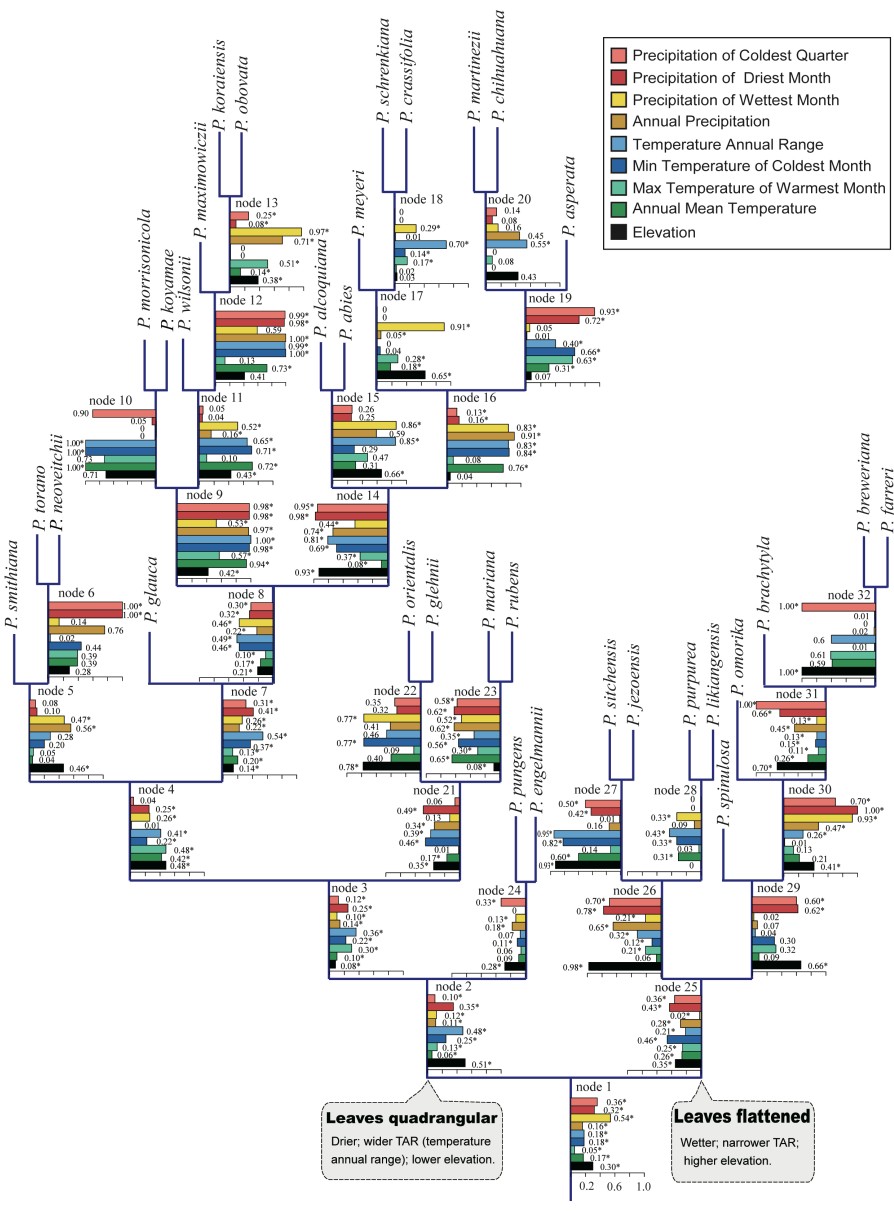

**Figure 3.** Divergence indices (scales range from 0-1) shown as histograms for elevation and for the 8 bioclimatic variables for each node of the morphology of *Picea* worldwide. *Indicates a significant difference of ecological features after Bonferroni correction ($P<0.0016$).





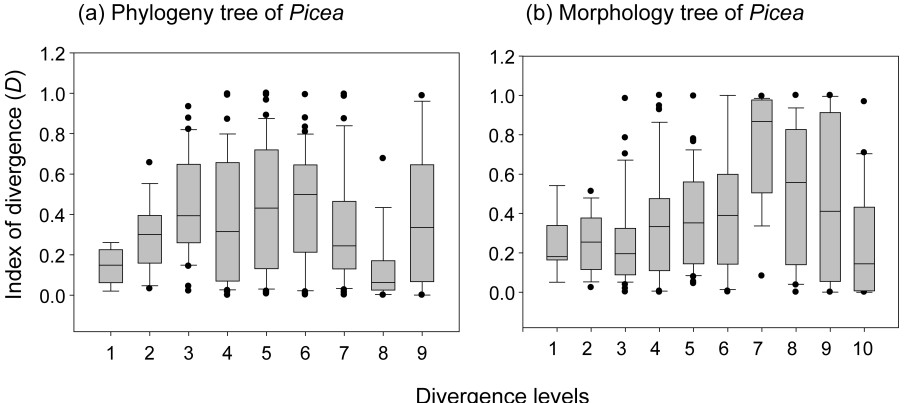

**Figure 4.** Box plots showing the index of divergence at each of the splitting levels in

the phylogenetic tree (a) and the morphological tree (b) of spruce species worldwide.

The central box in each box plot indicates the interquartile range and median, whereas

the whiskers show the 10th and 90th percentiles. Mean values marked with different

letters indicate a significant difference at $P<0.01$.