# Peer review of "Detecting climatically driven phylogenetic and morphological divergence among spruce (*Picea*) species worldwide"

_Biogeosciences, 2016_

## Referee Comment (RC1) · Anonymous Referee #1 · 29 Nov 2016

I read the manuscript "Detecting climatically driven phylogenetic and morphological divergence among spruce species (Picea) worldwide" with delight. The manuscript explored the relationship between climate and the phylogenetic and morphological divergence of Picea species in the Northern Hemisphere, based on 3388 georeferenced distribution sites. Temperature and precipitation parameters were the main driving factors for the primary divergence of spruce phylogeny and morphology, respectively. The climatic data extracted from current spruce locations captured the ecological divergence among spruces. These results suggested that the primary divergence of morphology and phylogeny among the spruces tended to be driven by different selective pressures. The data and methods are appropriate for this study, the manuscript is well

organized and presented. I found that the manuscript has a merit for publication in the journal Biogeosciences, pending on the authors can address my following concerns. My major concern is that if the climatic data used in this manuscript can represent the local climate of the distribution sites. The Worlclim dataset has been widely used in biogeographic studies. It can be used to surrogate the local climate in plain areas. However, it cannot represent the local climate in the mountainous regions because of the coarse resolution (about 1km). In the mountainous regions, 1 km distance may cover an elevational interval of hundreds of meters (and therefore introduce several degrees of difference in temperature). The authors need to discuss the caveat of using this dataset. Specific points Line 66: "Nearly 34 species" should be "Thirty-four species" Line 83: "niche conservatism" is not a process, but a pattern (result of the processes) Line 130: "Between 34 and 35 species" is conflict to the "34 species" (line 66) Line 132 (and other areas): "flora of China" should be "Flora of China" Line 145, 148, 152: delete "approximately" Line 159-166: set abbreviations for the climatic variables (and use the abbreviations afterwards) Line 216-224: move to the "Materials and Methods" section

---

## Referee Comment (RC2) · Anonymous Referee #2 · 13 Dec 2016

Wang et al., analyzed the relations between current climate and ecological (phylogenetic and morphological) divergence among spruce species at a global scale. The topic is suitable for Biogeosciences, but I do not think it is suitable for this special issue "Ecosystem processes and functioning across current and future dryness gradients in arid and semi-arid lands". The range of spruce (we could see in Figure 1) is not only limited in arid and semi-arid lands, but also covers a lot of other more wet regions. The main results of this paper is clear, that phylogenetic and morphological divergence are driven by different climate variables, i.e., temperature for phylo and precipitation for morpho. But I have several questions/comments, which need carefully revised by the authors. Firstly, the abstract is not well written. Too much information on methods

and results. Usually, we first need some background, importance of the study, come up with the question, and what we do, what we found, and finally the importance of our findings. Furthermore, some information in the abstract are repeated, e.g., line 30-34 and line 40-41. Other minor problems in Abstract include: bioclimatic or climatic (should be consistent here and other parts of this paper); global and northern hemisphere are different; there are ecological divergence, phylo divergence, morpho divergence and divergence, should be consistent or clearly defined; younger nodes are called remaining/terminal/end nodes/splits, should be consistent. Secondly, the use of current climate. The author also discussed this problem. As far as I know, there are not only current climate data in worldclim, but also paleoclimate. Although the paleoclimate there only date back to LGM, it still could reflect the climate situation for a longer time to some extent. I am wondering if this paleoclimate could be a better choice than current climate. Thirdly, the authors did PCA analysis and found that the first three axis could explain 75.67% of the variance, but the following analysis used 8 separate climate variables. I want to know why they choose these 8 variables, and not using the first three axis. Generally, 75% variance is OK. I guess the 4 temperature variables the author used are highly correlated, as well as the four precipitation variables. So I doubt the necessary to use so many climate variables. By the way, the numbers in the main text is not consistent with the numbers in table 1. For instance, the first axis explain 43.52% of the variance in Table 1, but 29.8% in the main text; other numbers are also wrong. In table 1, the first column, how did the authors choose the bold variables. I mean temperature seasonality is -0.928, and mean temperature of the coldest quarter is 0.946, higher than the AMT. The use of elevation is also questionable. The author at list did not discuss the effect of elevation in discussion. Other minor suggestions include: 1. The results do not need to be divided into 6 parts, I think the last 4 parts could be merged into 1. 2. Some logic in the text is not reasonable. For instance, in line 87, information before "thus" and after "thus", I don't think they are well connected; line 178-189, the sequence of these part is mess, line 188-189 should move to the front of the introduction of the SEEVA. The come up with several hypotheses in the

introduction also feel not well connected with the text there. Anyway, the authors need to carefully check this throughout in the text. 3. Line 148, mainland China and Taiwan? 4. Line 158-166, I am wondering if it's necessary to list all the climate variables here. 5. Line 349-350, how did the authors conclude like that? 6. Cannot or could not?

---

## Author Comment (AC2) · 10 Jan 2017

Major Comment:

[Comment 1] Wang et al., analyzed the relations between current climate and ecological (phylogenetic and morphological) divergence among spruce species at a global scale. The topic is suitable for Biogeosciences, but I do not think it is suitable for this special issue "Ecosystem processes and functioning across current and future dryness gradients in arid and semi-arid lands". The range of spruce (we could see in Figure 1) is not only limited in arid and semi-arid lands, but also covers a lot of other more wet regions. The main results of this paper are clear that phylogenetic and morphological divergence is driven by different climate variables, i.e., temperature for phylo and

precipitation for morpho. But I have several questions/comments, which need carefully revised by the authors.

[Response] The major reason why we submitted this MS to the special issue "Ecosystem processes and functioning across current and future dryness gradients in arid and semi-arid lands" is that most spruce species are very important taxa in arid and semi-arid lands worldwide. Detecting climatically driven phylogenetic and morphological divergence among spruce species worldwide would deepen the understanding of ecosystem processes and functioning in arid and semi-arid lands.

To address this point, in this response, we extracted the Aridity Index (AI) for each point from the Global Aridity Index (Global-Aridity) and the Global Potential Evapo-Transpiration (Global-PET) Geospatial Database (http://www.cgiar-csi.org/2010/04/134/), According the 1997 UNPE standard (Middleton Thomas, 1997) climate zone classification, 8 spruce species are in arid and Semi-Arid areas, 11 spruce species in Dry sub-humid areas, and 14 spruce species in humid areas. According to the scenario of global climate change, there would have severe and widespread droughts in the next 30-90 years over land areas resulting from either decreased precipitation and/or increased evaporation, and the significant increases in aridity do occur in many subtropical and adjacent humid regions [1, 2]. When overlapping the spruce sampling point to the future Aridity Changes Map (Fig. 1, 2 in this response), nearly all the spruce species whose original distribution in sub-humid and humid areas would subject to drought stress.

The Special issue "Ecosystem processes and functioning across current and future dryness gradients in arid and semi-arid lands" aims to provided platform for researches in plant species associations, plant distribution along environmental gradients, which is not only applicable for species distributed in arid and semi-arid areas, but also for the species subjected to aridity stress in future. Our findings would be helpful for management strategies and inform policy to climate change in future.

[Comment 2] Firstly, the abstract is not well written. There are too much information on methods and results. Usually, we first need some background, importance of the study, come up with the question, and what we do, what we found, and finally the importance of our findings. Furthermore, some information in the abstract are repeated, e.g., line 30-34 and line 40-41. Other minor problems in Abstract include: bioclimatic or climatic (should be consistent here and other parts of this paper); global and northern hemisphere are different; there are ecological divergence, phylo divergence, morpho divergence and divergence, should be consistent or clearly defined; younger nodes are called remaining/terminal/end nodes/splits, should be consistent.

[Response] Thank you for your comments. We will make these changes as suggested.

[Comment 3] The use of current climate: The author also discussed this problem. As far as I know, there are not only current climate data in worldclim, but also paleoclimate. Although the paleoclimate there only date back to LGM, it still could reflect the climate situation for a longer time to some extent. I am wondering if this paleoclimate could be a better choice than current climate.

[Response] Due to expansion and retreat occurred in the past, the present distribution of spruce is different from the distribution of the fossil locations. Thus, paleoclimate data does not necessarily match the present distribution. The 3388 data points of the 33 spruce species were sampled on present locations. Current climate data should be more appropriate to interpret current distribution pattern of spruce species.

[Comment 4] The authors did PCA analysis and found that the first three axes could explain 75.67

[Response] We actually ran the SEEVA by taking all the 16 climate factors into account. To illustrate the results briefly and clearly, we need to reduce the redundant variables. We focused on how mean value, extreme values of climate factors influence spruce divergence. In addition, The climatic variables must have higher divergence indices for the first split on the phylogeny and morphology of Picea, and relatively higher loading

on the five component axes. As a result, we mapped eight climate factors in the histograms on the phylogeny and morphology tree. Take an example, Min Temperature of Coldest Month and Mean Temperature of Coldest Quarter both have high loading on axis-1of PCA: 0.931 vs. 0.946, but the former has higher divergence indices than the latter (0.0764 vs. 05524 in the phylogeny and 0.18 vs. 0.08 in the morphology). We therefore illustrated the results of the former variable. Table-1 showed the eigenvalues, variance percentages, cumulative percentages and correlations of 19 bioclimatic factors but the rotated percentages were shown in the text. We will revise this inconsistency. Thanks. Spruce is elevation-sensitive. We selected elevation as a variable because it can demonstrate a direct view with respect to spruce divergence, which would be helpful to understand how topography influences spruce divergence.

Specific points:

[Comment 1]: The results do not need to be divided into 6 parts, I think the last 4 parts could be merged into 1.

[Response] Agree. We will reorganize the text. We think 3.3, 3.4 and 3.5 should be merged into one section.

[Comment 2]: Some logic in the text is not reasonable. For instance, in line 87, information before "thus" and after "thus", I don't think they are well connected; line 178-189, the sequence of these parts is mess, line 188-189 should move to the front of the introduction of the SEEVA. The come up with several hypotheses in the introduction also feel not well connected with the text there. Anyway, the authors need to carefully check this throughout in the text.

[Response] Thanks. We will check these during the revision stage.

[Comment 3] Line 148, mainland China and Taiwan?

[Response] Agree. "mainland China and Taiwan" should be more formal. We will check these.

[Comment 4] Line 158-166, I am wondering if it's necessary to list all the climate variables here.

[Response] Agree. A full list of climate variables has been shown in Table 1. We will check these during the revision stage.

[Comment 5] Line 349-350, how did the authors conclude like that? 6. Cannot or could not?

[Response] This paragraph highlighted the exceptions observed for a few sister groups or species in the phylogenetic tree to the overall pattern. We explained these exceptions as a result of geographical isolation and the limitation of the selected climate parameters that do not adequately describe the climatic determinants of spruce distributions.

Reference

Budantsev LY (1994) The Fossil Flora of the Paleogene Climatic Optimum in North Eastern Asia, Springer Berlin Heidelberg.

Dai A (2012) Increasing drought under global warming in observations and models. Nature Climate Change, 3, 52-58.

Farjón A (2001) World Checklist and Bibliography of Conifers (Second edn.) England, Cambridge University Press.

Giesecke T (2004) The Holocene Spread of Spruce in Scandinavia.

Greve P, Seneviratne SI (2015) Assessment of future changes in water availability and aridity. Geophys Res Lett, 42, 5493-5499.

Hang S (2002) Evolution of Arctic-Tertiary flora in Himalayan-Hengduan mountains. Acta Botanica Yunnanica, 24, 671-688.

Kullman L (1995) New and firm evidence for Mid-Holocene appearance of Picea abies

in the Scandes Mountains, Sweden. Journal of Ecology, 83, 439-447.

Liu JQ, Gao TG, Chen ZD, Lu AM (2002) Molecular phylogeny and biogeography of the Qinghai-Tibet Plateau endemic Nannoglottis (Asteraceae). Molecular Phylogenetics Evolution, 23, 307-325.

Ran JH, Wei XX, Wang XQ (2006) Molecular phylogeny and biogeography of Picea (Pinaceae): implications for phylogeographical studies using cytoplasmic haplotypes. Mol Phylogenet Evol, 41, 405-419.

Spribille T, Chytry M (2002) Vegetation surveys in the circumboreal coniferous forests: A review. Folia Geobotanica, 37, 365-382.

Struwe L, Smouse PE, Heiberg E, Haag S, Lathrop RG (2011) Spatial evolutionary and ecological vicariance analysis (SEEVA), a novel approach to biogeography and speciation research, with an example from Brazilian Gentianaceae. Journal of Biogeography, 38, 1841-1854.

Wu S, Yang YP, Fei Y (1995) On the flora of the alpine region in the Qinghai-Xizang (Tibet) Plateau, China. Acta Botanica Yunnanica.

[Figure]

Figure 1. The locations of sampling point in the study at different climate zone. The background image was the map of Global Aridity Index which obtained online (http://www.cgiar-csi.org)by the CGIAR-CSI with the support of the International Center for Tropical Agriculture (CIAT).

**Fig. 1.** The locations of sampling point in the study at different climate zone

[Figure]

Figure 2. The locations of sampling point in Aridity changes within the 21st century. The background image was the map of Changes in P–Rn∕λ comparing present-day (1980–2000) and future climate (2080–2100) following the RCP8.5 pathway.(P, precipitation; Rn, net radiation;  , latent heat of vaporization).(Greve &  Seneviratne, 2015).

**Fig. 2.** The locations of sampling point in Aridity changes within the 21st century.

---

## Author Response (AR1)

**Responses to Referee #1**

**Major Comment:**

*[Comment]* I read the manuscript "Detecting climatically driven phylogenetic and morphological divergence among spruce species (*Picea*) worldwide" with delight. The manuscript explored the relationship between climate and the phylogenetic and morphological divergence of *Picea* species in the Northern Hemisphere, based on 3388 georeferenced distribution sites. Temperature and precipitation parameters were the main driving factors for the primary divergence of spruce phylogeny and morphology, respectively. The climatic data extracted from current spruce locations captured the ecological divergence among spruces. These results suggested that the primary divergence of morphology and phylogeny among the spruces tended to be driven by different selective pressures. The data and methods are appropriate for this study; the manuscript is well organized and presented. I found that the manuscript has a merit for publication in the journal Biogeosciences, pending on the authors can address my following concerns. My major concern is that if the climatic data used in this manuscript can represent the local climate of the distribution sites. The Worlclim dataset has been widely used in biogeographic studies. It can be used to surrogate the local climate in plain areas. However, it cannot represent the local climate in the mountainous regions because of the coarse resolution (about 1km). In the mountainous regions, 1 km distance may cover an elevational interval of hundreds of meters (and therefore introduce several degrees of difference in temperature). The authors need to discuss the caveat of using this dataset.

[Response] This is a good question. The coarse resolution (about 1km) of climate data from The Worlclim dataset would likely weaken the potential to interpret spruce distribution and divergence. Discussion of the caveat of using this dataset is absolutely needed and will be done when we get the chance to revise this MS. Thank you!

Nevertheless, we have the confidence that the climate data from The WorlClim dataset used in this study is suitable for interpreting the overall pattern, i.e., the first several splits that represent "the primary trigger" that led to the divergence of among spruce, which are the major findings of this study. As we can see from Fig.1 (a, b), instead of elevation gradient, the geographical distribution of both the three phylogenetic clades and the morphological groups (quadrangular leaves versus flattened leaves) is largely determined by horizontal gradients (latitude and longitude). Specifically, clade-1 is a Eurasian clade and clade-2 is a North American clade, while clade-3 is an Asian clade with only one North American species. As for as the morphological groups, spruces flattened leaves tend to occur in eastern Asia and the beach area of the northern America, while spruce with quadrangular leaves distribute in the rest part of the whole distribution range. Given this base, the 1km-resolution of climate data we used in this study should be robust to interpret this large scale pattern.

We confess this dataset may give rise to some uncertainties in the context of the detection of some subtle variation such as within-species variation or among elevation-sensitive species. In this case, although the splits at the terminal nodes are between species, that is to say that we don't have any within-species variation, the caveat with respect to the dataset must be discussed. Further works that focus on the driving force underlying the variation of within-species or among elevation-sensitive species should use high resolution climate data.

**Specific points:**

*[Comment 1]* Line 66: "Nearly 34 species" should be "Thirty-four species"

[Response] We will make this change as suggested.

*[Comment 2]* Line 83: "niche conservatism" is not a process, but a pattern (result of the processes)

[Response] Agree! We will change the "process" to "pattern" in the revised manuscript.

*[Comment 3]* Line 130: "Between 34 and 35 species" is conflict to the "34 species" (line 66) Line 132 (and other areas): "flora of China" should be "Flora of China"

[Response] Thank you. We will make this change as suggested.

*[Comment 4]* Line 145, 148, 152: delete "approximately"

[Response] Thank you. We will make this change as suggested.

*[Comment 5]* Line 159-166: set abbreviations for the climatic variables (and use the abbreviations afterwards)

[Response] Thank you. We will make this change as suggested.

*[Comment 6]* Line 216-224: move to the "Materials and Methods" section

**Reference**

Harris I, Jones PD, Osborn TJ, Lister DH (2014) Updated high-resolution grids of monthly climatic observations - the CRU TS3.10 Dataset. International Journal of Climatology, **34**, 623-642.

Hijmans RJ, Cameron SE, Parra JL, Jones PG, Jarvis A (2005) Very high resolution interpolated climate surfaces for global land areas. International Journal of Climatology, **25**, 1965-1978.

Kriticos DJ, Webber BL, Leriche A, Ota N, Macadam I, Bathols J, Scott JK (2012) CliMond: global high-resolution historical and future scenario climate surfaces for bioclimatic modelling. Methods in Ecology and Evolution, **3**, 53-64.

**Responses to Referee #2**

**Major Comment:**

**[Comment 1]** Wang et al., analyzed the relations between current climate and ecological (phylogenetic and morphological) divergence among spruce species at a global scale. The topic is suitable for Biogeosciences, but I do not think it is suitable for this special issue "Ecosystem processes and functioning across current and future dryness gradients in arid and semi-arid lands". The range of spruce (we could see in Figure 1) is not only limited in arid and semi-arid lands, but also covers a lot of other more wet regions. The main results of this paper are clear that phylogenetic and morphological divergence is driven by different climate variables, i.e., temperature for phylo and precipitation for morpho. But I have several questions/comments, which need carefully revised by the authors.

**[Response]** The major reason why we submitted this MS to the special issue "Ecosystem processes and functioning across current and future dryness gradients in arid and semi-arid lands" is that most spruce species are very important taxa in arid and semi-arid lands worldwide. Detecting climatically driven phylogenetic and morphological divergence among spruce species worldwide would deepen the understanding of ecosystem processes and functioning in arid and semi-arid lands.

To address this point, in this response, we extracted the Aridity Index (AI) for each point from the Global Aridity Index (Global-Aridity) and the Global Potential Evapo-Transpiration (Global-PET) Geospatial Database (http://www.cgiar-csi.org/2010/04/134/), According the 1997 UNPE standard (Middleton& Thomas, 1997) climate zone classification, 8 spruce species are in arid and Semi-Arid areas, 11 spruce species in Dry sub-humid areas, and 14 spruce species in humid areas. According to the scenario of global climate change, there would have severe and widespread droughts in the next 30-90 years over land areas resulting from either decreased precipitation and/or increased evaporation, and the significant increases in aridity do occur in many subtropical and adjacent humid regions [1, 2]. When overlapping the spruce sampling point to the future Aridity Changes Map (Fig. 1, 2 in this response), nearly all the spruce species whose original distribution in sub-humid and humid areas would subject to drought stress.

The Special issue "Ecosystem processes and functioning across current and future dryness gradients in arid and semi-arid lands" aims to provided platform for researches in plant species associations, plant distribution along environmental gradients, which is not only applicable for species distributed in arid and semi-arid areas, but also for the species subjected to aridity stress in future. Our findings would be helpful for management strategies and inform policy to climate change in future.

[Figure]

Fig. 1. The locations of sampling point in the study at different climate zone. The background image was the map of Global Aridity Index which obtained online (http://www.cgiar-csi.org)by the CGIAR-CSI with the support of the International Center for Tropical Agriculture (CIAT).

[Figure]

Fig. 2. The locations of sampling point in Aridity changes within the 21st century. The background image was the map of Changes in P–R$n/\lambda$ comparing present-day (1980–2000) and future climate (2080–2100) following the RCP8.5 pathway.(P, precipitation; Rn, net radiation; $\lambda$, latent heat of vaporization).[1].

*[Comment 2]* Firstly, the abstract is not well written. There are too much information on methods and results. Usually, we first need some background, importance of the study, come up with the question, and what we do, what we found, and finally the importance of our findings. Furthermore, some information in the abstract are repeated, e.g., line 30-34 and line 40-41. Other minor problems in Abstract include: bioclimatic or cli- matic (should be consistent here and other parts of this paper); global and northern hemisphere are different; there are ecological divergence, phylo divergence, morpho divergence and divergence, should be consistent or clearly defined; younger nodes are called remaining/terminal/end nodes/splits, should be consistent.

**[Response]** Thank you for your comments. We will make these changes as suggested.

*[Comment 3]* The use of current climate: The author also discussed this problem. As far as I know, there are not only current climate data in worldclim, but also paleoclimate. Although the paleoclimate there only date back to LGM, it still could reflect the climate situation for a longer time to some extent. I am wondering if this paleoclimate could be a better choice than current climate.

**[Response]** Due to expansion and retreat occurred in the past, the present distribution of spruce is different from the distribution of the fossil locations. Thus, paleoclimate data does not necessarily match the present distribution. The 3388 data points of the 33 spruce species were sampled on present locations. Current climate data should be more appropriate to interpret current distribution pattern of spruce species.

*[Comment 4]* The authors did PCA analysis and found that the first three axes could explain 75.67% of the variance, but the following analysis used 8 separate climate variables. I want to know why they choose these 8 variables, and not using the first three axes. Generally, 75% variance is OK. I guess the 4 temperature variables the author used are highly correlated, as well as the four precipitation variables. So I doubt the necessary to use so many climate variables. By the way, the numbers in the main text is not consistent with the numbers in table 1. For instance, the first axis explains 43.52% of the variance in Table 1, but 29.8% in the main text; other numbers are also wrong. In table 1, the first column, how did the authors choose the bold variables. I mean temperature seasonality is -0.928, and mean temperature of the coldest quarter is 0.946, higher than the AMT. The use of elevation is also questionable. The author at list did not discuss the effect of elevation in discussion.

**[Response]** We actually ran the SEEVA by taking all the 16 climate factors into account. To illustrate the results briefly and clearly, we need to reduce the redundant variables. We focused on how mean value, extreme values of climate factors influence spruce divergence. In addition, The climatic variables must have higher divergence indices for the first split on the phylogeny and morphology of *Picea*, and relatively higher loading on the five component axes. As a result, we mapped eight climate factors in the histograms on the phylogeny and morphology tree. Take an example, Min Temperature of Coldest Month and Mean Temperature of Coldest Quarter both have high loading on axis-1of PCA: 0.931 vs. 0.946, but the former has higher divergence indices than the latter (0.0764 vs. 05524 in the phylogeny and 0.18 vs. 0.08 in the morphology). We therefore illustrated the results of the former variable.

Table-1 showed the eigenvalues, variance percentages, cumulative percentages and correlations of 19 bioclimatic factors but the rotated percentages were shown in the text. We will revise this inconsistency. Thanks.

Spruce is elevation-sensitive. We selected elevation as a variable because it can demonstrate a direct view with respect to spruce divergence, which would be helpful to understand how topography influences spruce divergence.

**Specific points:**

*[Comment 1]*: The results do not need to be divided into 6 parts, I think the last 4 parts could be merged into 1.

**[Response]** Agree. We will reorganize the text. We think 3.3, 3.4 and 3.5 should be merged into one section.

*[Comment 2]:* Some logic in the text is not reasonable. For instance, in line 87, information before "thus" and after "thus", I don't think they are well connected; line 178-189, the sequence of these parts is mess, line 188-189 should move to the front of the introduction of the SEEVA. The come up with several hypotheses in the introduction also feel not well connected with the text there. Anyway, the authors need to carefully check this throughout in the text.

**[Response]** Thanks. We will check these during the revision stage.

*[Comment 3]* Line 148, mainland China and Taiwan?

**[Response]** Agree. "mainland China and Taiwan" should be more formal. We will check these.

*[Comment 4]* Line 158-166, I am wondering if it's necessary to list all the climate variables here.

**[Response]** Agree. A full list of climate variables has been shown in Table 1. We will check these during the revision stage.

*[Comment 5]* Line 349-350, how did the authors conclude like that? 6. Cannot or could not?

**[Response]** This paragraph highlighted the exceptions observed for a few sister groups or species in the phylogenetic tree to the overall pattern. We explained these exceptions as a result of geographical isolation and the limitation of the selected climate parameters that do not adequately describe the climatic determinants of spruce distributions.

**A list of all relevant changes made in the manuscript**

- We have simplified and specified abstract.

- We add a describe of climate zone of spruce species distributed in the method section and discussion section.

- We Added a statement of significance of the findings in relation to climate change to abstract, and added more discussions about significance and implications of the findings of this research in relation to future climate change.

- We have improved the English writing and sent the manuscript to a language service, made the paper more formal.

[revised manuscript text omitted]

批注 [E1]: Add a hyphen between 'Clade' .

[Figure]

批注 [E2]: Changed "temperature annual range" to "annual temperature range".

---

## Editor Decision (ED1)

Journal: BG
Title: Detecting climatically driven phylogenetic and morphological divergence among spruce
species (Picea) worldwide
Author(s): Guo-Hong Wang et al.
MS No.: bg-2016-465
MS Type: Research article
Special Issue: Ecosystem processes and functioning across current and future dryness gradients
in arid and semi-arid lands

**General comments:**

The authors of the manuscript (bg-2016-465) addressed the reviewers' issues clearly and
answered our questions well. The manuscript has been largely improved and reaches the standard
of BG.  Therefore, I decide a minor revision.  However, although authors clearly showed point-
by-point revisions on the basis of reviewers' and my comments by a marked-up copy of the
revised manuscript, I would request authors to provide a point-by-point response list to each
reviewer's and my comments, showing how and where the revisions are in the revised
manuscript. In addition, I have more minor comments as listed below for authors to respond.

**Specific comments:**

1. Line 13-14, what do you mean by 'at 31 nodes' and 'at 32 nodes'?
2. L16, add complete name of Dmax because it is mentioned first time.
3. Please add some new references published in year 2015 and 2016 in introduction and
   discussion.
4. Line 158-159, why you use P value less than 0.0016? I suggest a consistent significant
   level of 0.05 or 0.01 throughout text. Besides, I would like to have P value when you talk
   about statistical significance.
5. Line 160, rewrite '$\alpha=0.05/31$ or $32\approx0.0016$'.
6. L220-222, move this statement to discussion.
7. L225, use specific p value.
8. L228-248, keep verb tense consistent throughout text. Similarly, check the tense
   throughout text.
9. L297, associated with
10. L299, what is the first hypothesis? You may underscore in the introduction or brief it
    here. Authors did not test the so-called second hypothesis that authors mentioned in the
    introduction. Please test the second hypothesis.
11. L432, please specify the statement 'Our hypotheses are largely verified......'. Which
    hypothesis you specifically referred to?

---

## Author Response (AR2)

**Responses to the Associate Editor (major revision)**

Comments to the Author:

**[General comments]** This paper elucidated the relationship between climate and the phylogenetic and morphological divergence of global spruces (Picea) in the Northern Hemisphere. The authors found that temperature parameters and precipitation parameters tended to be the main driving factors for the primary divergence of spruce phylogeny and morphology, respectively. The primary divergence of morphology and phylogeny among the investigated spruces at 3388 sites tended to be driven by different selective pressures.

I think this paper is scientifically significant. The subject of the paper fall within the scope of the BG special issue (Ecosystem processes and functioning across current and future dryness gradients in arid and semi-arid lands).

It is concise, well structured, and well written. The manuscript has a merit for publication in the SI (Ecosystem processes and functioning across current and future dryness gradients in arid and semi-arid lands). However, the manuscript can potentially be improved largely on the basis of referees' comments and authors responses to the comments.   Therefore, I decide a major revision.

**[Response]**

We have finished the major revision by taking all the comments into account.

My specific comments:

**[Comment 1]** Please simplify and specify abstract.

**[Response]** Abstract has been rewritten.

**[Comment 2]** Add climatic zone distribution to the methods section.

**[Response] Done** (L108-L109).

**[Comment 3]** Add a statement of significance of the findings in this paper in relation to climate change to abstract.

Discuss significance and implications of the findings of this research in relation to future climate change.

**[Response]** This is an important point. We highlighted the significance and implications of our findings in relation to future climate change (L19-L22, L429-L443).

**Responses to the Associate Editor (minor revision)**

**[General comments]** The authors of the manuscript (bg-2016-465) addressed the reviewers' issues clearly and answered our questions well. The manuscript has been largely improved and reaches the standard of BG. Therefore,

I decide a minor revision. However, although authors clearly showed point-by-point revisions on the basis of reviewers' and my comments by a marked-up copy of the revised manuscript, I would request authors to provide a point-by-point response list to each reviewer's and my comments, showing how and where the revisions are in the revised manuscript. In addition, I have more minor comments as listed below for authors to respond.

**[Response]** We have made a point-by-point response list to each reviewer's and the Associate Editor comments. All the comments have been taken into account in the context of revision process.

**Specific comments:**

**[Comment 1]** Line 13-14, what do you mean by 'at 31 nodes' and 'at 32 nodes'?

**[Response]** The phylogenetic tree and the morphological tree include 31 nodes and 32 nodes respectively. Nine comparisons (nine environmental factors) were conducted for each node. Accordingly, a total of 279 comparisons and 288 comparisons were conducted for the phylogenetic tree and the morphological tree, respectively.

**[Comment 2]** L16, add complete name of because it is mentioned first time.
**[Response]** Agree. The $D_{max}$ in L16 was replaced by the maximum $D$.

**[Comment 3]** Please add some new references published in year 2015 and 2016 in introduction and discussion.
**[Response] Agree.** We have retrieved added the latest and related publications in this revision.

**[Comment 4]** Line 158-159, why you use P value less than 0.0016? I suggest a consistent significant level of 0.05 or 0.01 throughout text. Besides, I would like to have P value when you talk about statistical significance.
**[Response]** P value less than 0.0016 was a result of Bonferroni correction, i.e., '$\alpha=0.05/(31$ or $32)\approx0.0016$', because 31 and 32 independent tests were conducted for each of the climatic variables. Thus, a $P$-value less than 0.0016 indicated a significant difference in the ecological features for splits at a given node.

**[Comment 5]** Line 160, rewrite '$\alpha=0.05/31$ or $32\approx0.0016$'.
**[Response]** '$\alpha=0.05/31$ or $32\approx0.0016$' was replaced by '$\alpha=0.05/(31$ or $32)\approx0.0016$' in L166.

**[Comment 6]** L220-222, move this statement to discussion.
**[Response]** Done.

**[Comment 7]** L225, use specific p value.
**[Response]** Done.

**[Comment 8]** L228-248, keep verb tense consistent throughout text. Similarly, check the tense throughout text.
**[Response]** Done.

**[Comment 9]** L297, associated with
**[Response]** Done.

**[Comment 10]** L299, what is the first hypothesis? You may underscore in the introduction or brief it here. Authors did not test the so-called second hypothesis that authors mentioned in the introduction. Please test the second hypothesis.
**[Response]** The first hypothesis was presented in L73-L78. We reiterated it briefly in L309- L310. The second one was in L88-L90. The test to this hypothesis was addressed in 4.4 (L410-L428). We underscored the test of the second hypothesis in L426.

**[Comment 11]** L450, please specify the statement 'Our hypotheses are largely verified……'. Which hypothesis you specifically referred to?
**[Response]** Agree. In L447-L453 in this revised version, we specified the test to our hypothesis by highlighting the major findings from this study.

[revised manuscript text omitted]
 mean temperature (AMT), mean temperature diurnal range (MTDR), isothermality (ISO), temperature seasonality (TS), maximum temperature of the warmest month (MTWM), minimum temperature of coldest month (MTCM), annual temperature range (ATR), mean temperature of the wettest quarter (MTWQ), mean temperature of the driest quarter (MTDQ), mean temperature of the warmest quarter (MTWQ), mean temperature of the coldest quarter (MTCQ), mean annual precipitation (AP), precipitation of the wettest month (PWM), precipitation of the driest month (PDM), precipitation seasonality (PS), precipitation of the wettest quarter (PWQ), precipitation of the driest quarter (PDQ), precipitation of the warmest quarter (PWQ) and precipitation of the coldest quarter (PCQ). The values of each climate variable at each site were extracted using the software QGIS (http://qgis.osgeo.org), and the final data were exported to an Excel worksheet for subsequent analysis. A factor analysis was conducted to eliminate the redundant climatic variables, and a principal component analysis (PCA) of the climatic variables was performed using the SPSS statistical package (SPSS, Chicago, IL, USA), Therefore, we selected eight bioclimatic variables for subsequent analysis, including four temperature variables (annual mean temperatureMAT, minimum temperature of the coldest monthMTCM, maximum temperature of the warmest monthMTWM and annual temperature rangeATR) and four precipitation variables (annual precipitationAP, precipitation of the wettest monthPWM, precipitation of the driest monthPDM and precipitation of the coldest quarterPCQ). 
[revised manuscript text omitted]